# Answer Me if You Can:
# Debiasing Video Question Answering
# via Answering Unanswerable Questions

## Abstract

Video Question Answering (VideoQA) is a task to predict a correct answer given a question-video pair. Recent studies have shown that most VideoQA models rely on spurious correlations induced by various biases when predicting an answer. For instance, VideoQA models tend to predict 'two' as an answer without considering the video if a question starts with "How many" since the majority of answers to such type of questions are 'two'. In causal inference, such bias (*question type*), which simultaneously affects the input $X$ (*How many...*) and the answer $Y$ (*two*), is referred to as a confounder $Z$ that hinders a model from learning the true relationship between the input and the answer. The effect of the confounders $Z$ can be removed with a causal intervention $P(Y|do(X))$ when $Z$ is observed. However, there exist many unobserved confounders affecting questions and videos, *e.g.*, dataset bias induced by annotators who mainly focus on human activities and salient objects resulting in a spurious correlation between videos and questions. To address this problem, we propose a novel framework that learns unobserved confounders by capturing the bias using *unanswerable* questions, which refers to an artificially constructed VQA sample with a video and a question from two different samples, and leverages the confounders for debiasing a VQA model through causal intervention. We demonstrate that our confounders successfully capture the dataset bias by investigating which part in a video or question that confounders pay attention to. Our experiments on multiple VideoQA benchmark datasets show the effectiveness of the proposed debiasing framework, resulting in an even larger performance gap compared to biased models under the distribution shift.

## 1 Introduction

Video Question Answering (VideoQA) task is a multi-modal understanding task to find the correct answer given a question-video pair, which requires an understanding of both vision and text modalities along with causal reasoning. However, recent studies (Ramakrishnan et al., 2018; Cadene et al., 2019) point out that the success of the VideoQA models is due to its reliance on *spurious correlations* caused by bias instead of reasonable inference for answer prediction. In other words, the models concentrate on the co-occurrence between the question (or video) and the answer based on the dataset statistics and tend to simply predict the frequent answers. For instance, given a question that starts with "How many", a biased VideoQA model often blindly predicts 'two' as an answer as depicted in Fig. 1b. Fig. 1a illustrates the statistics of MSVD-QA dataset, showing that the majority of the answer to the "How many" questions are 'two'. In this case, 'question type' is acting as a bias simultaneously influencing the input question-video pair and the answer, which hinders the model from learning a true relationship between the input and the answer.

In causal inference (Glymour et al., 2016), such variable, *e.g.*, question type, affecting both the input $X$ and the answer $Y$ is called a confounder $Z$, which interrupts finding a true causal relationship between $X$ and $Y$. The causal intervention $P(Y|do(X))$ intentionally cuts off the relation between $X$ and $Z$ via *do*-calculus, which is also called 'deconfounding', to remove the effect of the con-

founders[1]. Nevertheless, $Z$ should be predefined to apply causal intervention but most confounders are unobserved in the dataset and hard to be applied to the causal intervention.

Therefore, we introduce *learnable confounder queries* and train them to capture the bias, and leverage the learned confounders for debiasing through the causal intervention. To achieve this, we force the model to answer the *unanswerable* question. An unanswerable question refers to an artificially constructed VQA sample with a video and a question from two different samples in a mini-batch, along with an answer that corresponds to either the video or the question. When a model answers the unanswerable question, the model inevitably learns the bias of the specific modality that corresponds to the answer since another modality is randomly sampled and irrelevant to the answer.

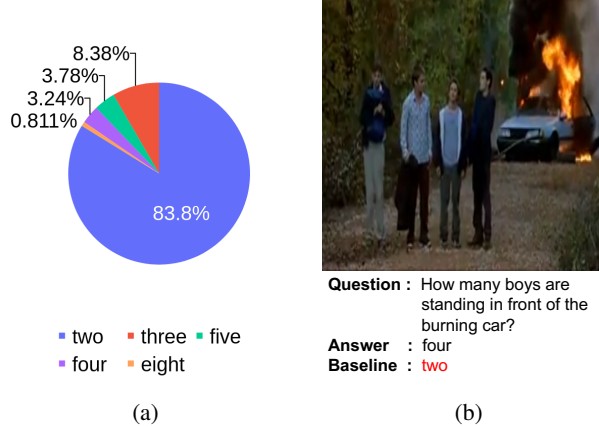

Figure 1: **Dataset statistics of MSVD dataset and an example of the biased answer.** (Left) The majority of answers to the "How many" questions are 'two'. (Right) The model outputs the biased answer 'two' instead of the right answer 'four'.

To summarize, we propose a novel framework Debiasing a **V**ide**o** Questi**o**n Answe**r**ing Mo**d**el by Answering Unanswerable **Q**uestions (VoidQ) with causal inference. In order to apply the causal intervention $P(Y|do(X))$, we introduce learnable confounder queries. The proposed confounder queries are trained to capture the bias by answering the unanswerable questions. Our framework leverages the confounder queries and their outputs to debias our VQA model via causal intervention. We validate our models on three benchmark VideoQA datasets (TGIF-QA, MSRVTT-QA, and MSVD-QA) and demonstrate the effectiveness of our debiasing strategy. Also, ablation studies reveal that the performance gap between conventional biased models and the proposed model gets larger when the training and test distribution significantly differ, supporting the improved generalization ability by the proposed approach. Lastly, visualization of confounders via our variant of Grad-CAM shows that the learned confounder queries adequately debias a VQA model by taking into account falsely correlated keywords in questions or salient regions in videos.

To sum up, our **contributions** are as follows:

- We propose a novel debiasing framework for VideoQA model to predict correct answers based on causal inference by removing the effect of the confounders.
- We also present a training scheme encouraging the learnable confounder queries to capture the bias by forcing the model to answer the unanswerable questions.
- Our extensive experiments demonstrate that the proposed framework outperforms previous models on various benchmark datasets, even with a larger margin under the distribution shift where the biased models suffer significant performance degradation from.
- We verify that our confounders successfully capture the dataset bias by investigating which parts in a video or which words in a question are utilized by confounder queries to correct the predictions of a VQA model.

## 2 RELATED WORK

**Video Question Answering (VideoQA).** VideoQA is a task to infer a correct answer, given a video and a question. While models for the VisualQA task focuses on spatial information of an image (Antol et al., 2015; Yang et al., 2016), VideoQA requires reasoning over both temporal and spatial dynamics, making it a more challenging task. Previous works have applied spatio-temporal contextual attention to various scenarios (Jang et al., 2017; Xiao et al., 2022; Zhao et al., 2017).

---

[1]Additional descriptions about the causal inference are in Sec. 3.1

Another line of research has proposed the end-to-end pretrained models on the large-scale dataset to improve the performance of various downstream tasks including the VideoQA. Fu et al. (2021) builds a additional cross-modal encoder as well as the video and text encoder and Wang et al. (2022) introduce a token rolling operation to efficiently perform the temporal attention on the cross-modal encoder. However, existing models still suffer from dataset bias (Ramakrishnan et al., 2018; Cadene et al., 2019). Therefore, in this paper, we propose a debiasing framework for VideoQA to improve generalization power by reducing the dataset bias even under a distribution shift.

**Debiasing from the biased model.**  The first approach to alleviate the bias is to directly augment the dataset to remove statistical 'hints' and enlarge the size and diversity of the training set. Gokhale et al. (2020), Chen et al. (2020), and Kil et al. (2021) propose to augment input images or questions to generate counterfactual or paraphrased QA pairs. On the other hand, there exist attempts to learn the text or image bias through additional branches by training the branch only with a single modality. Outputs from the biased branches are then utilized to debias training (Ramakrishnan et al., 2018; Zhang et al., 2021; Cadene et al., 2019). However, these approaches require manually designed heuristic rules or separate branches to capture bias from a specific modality. We propose a novel unified debiasing framework that leverages the unanswerable questions, allowing the learnable confounder queries to capture the biases related to both modalities without any heuristics.

**Causal Inference.**  Causal inference, a method to find the *true* effect of a particular variable on a target variable without being interrupted by any other variables, is being widely adopted in Visual QA tasks. Previous works proposed augmenting the data to remove unwanted effects of a specific variable by utilizing the Structural Causal Model (SCM) (Glymour et al., 2016) which defines causal relationships between variables. Specifically, existing works generate counterfactual samples (Tang et al., 2020; Abbasnejad et al., 2020; Yue et al., 2021) or negative samples (Wen et al., 2021; Teney et al., 2020) to measure and remove an effect of a specific confounding variable. Besides, causal intervention is also used to directly remove the effects of a predefined confounding variable that hinders proper reasoning. Unfortunately, since the confounders are usually *unobserved*, most existing methods manually predefine the confounder sets as object classes (Zhang et al., 2020) or verb-centered relation tuples from caption data (Nan et al., 2021) in order to conduct the causal intervention. Unlike these works, we directly train the confounder queries so that they can capture various types of bias, instead of manually predefining what confounder should be.

## 3 METHOD

Bias misleads the model to become reliant on spurious correlations, resulting in poor generalization ability. Therefore, in this section, we propose a novel framework Debiasing a **V**ide**o** Quest**i**on Answer**i**ng Mo**d**el by Answering Unanswerable **Q**uestions (VoidQ) with causal inference. Firstly, we briefly revisit the basic concepts of the VideoQA and causal inference. We then present the debiasing framework with learnable confounder queries based on the causal intervention. Finally, we introduce the training objective using unanswerable questions to let the confounder queries learn the bias.

### 3.1 PRELIMINARIES

**VideoQA.**  VideoQA is a task to predict the answer $\hat{Y}$ given a question-video pair $X = (x_q, x_v)$. There are two types of tasks in the VideoQA: multi-choice question answering (MCQA) and open-ended question answering (OEQA). For the MCQA, the model predicts the answer among five options in general. Each option is concatenated with the question and the model calculates the similarities between each concatenated text and the video to output the final prediction. In the OEQA setting, the task is mostly converted to the classification task to predict the correct answer among the predefined global vocab-set containing all the candidate answers. For simplicity, we will explain concepts only with OEQA. Further details including MCQA are in the supplement. The prediction $\hat{Y}$ under the OEQA setting given a pair $X = (x_q, x_v)$ can be written as:

$$\hat{Y} = P(Y|X) = h(f(X)), \tag{1}$$

where $f$ is a feature encoder and $h$ is a classifier.

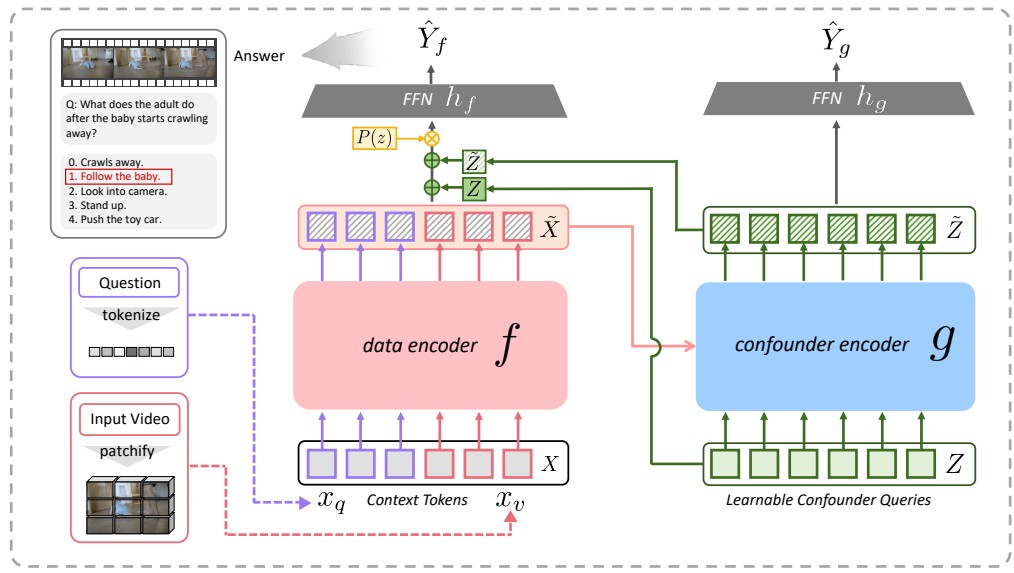

Figure 3: **VoidQ Architecture.** We construct tokenized text tokens $x_q$ and patchified video tokens $x_v$ from input question and video, then concatenate them to form $X$ and feed them into the data encoder $f$. Learnable confounder queries $Z$ are fed to the confounder encoder $g$, which are cross-attended with output features of $f$, *i.e.*, $\tilde{X}$. $Z$ is trained to learn the dataset bias by minimizing $\mathcal{L}_{\text{confounder}}$ between the ground truth and a biased prediction $\hat{Y}_g$ generated from $\tilde{Z}$ through $h_g$. Causal intervention utilizing learned confounders $Z$ and $\tilde{Z}$ is applied to generate the final debiased prediction $\tilde{Y}_f$ from $\tilde{X}$ through $h_f$.

**Causal Inference.** To train $f$ and $h$ in Eq. 1, most previous approaches (Fan et al., 2019; Gao et al., 2018; Jiang et al., 2020; Jiang & Han, 2020; Le et al., 2020) have adopted the standard cross entropy (CE) loss as:

$$\min_{f,h} \text{CE}(\hat{Y}, Y). \tag{2}$$

Eq. 2 aims simply to minimize CE between the ground-truth $Y$ and the prediction $\hat{Y}$, in which $f$ and $h$ naturally learn the spurious correlation between $X$ and $Y$.

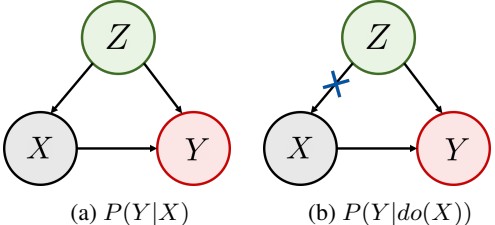

(a) $P(Y|X)$  (b) $P(Y|do(X))$

Figure 2: **Causal Inference.** $X$ is a cause, $Y$ is an effect, and $Z$ is a set of confounders.

To address this problem, some recent works (Tang et al., 2020; Zhang et al., 2020; Niu et al., 2021) tried applying causal inference (Glymour et al., 2016) to alleviate the bias. As shown in Fig. 2a, conventional approaches have calculated the likelihood $P(Y|X)$ to predict the answer as:

$$P(Y|X) = \sum_{z \in Z} P(Y|X, z)P(z|X). \tag{3}$$

On the other hand, the predicted answer with causal intervention $P(Y|do(X))$ where the connection between $X$ and $Z$ is cut-off is defined as:

$$P(Y|do(X)) = \sum_{z \in Z} P(Y|X, z)P(z). \tag{4}$$

Unlike in Eq. 3, the causal intervention removes the relation between $X$ and $Z$, allowing the model to reason about the real causation of $X$ to $Y$ by considering the prior $P(z)$ instead of $P(z|X)$ in Eq. 4. Note that a set of confounders $Z$ must be known in advance to calculate Eq. 4.

## 3.2 OVERALL ARCHITECTURE

VoidQ consists of two encoders. A data encoder $f$ follows the Transformer (Vaswani et al., 2017) encoder based on the self-attention mechanism. Unlike $f$, a confounder encoder $g$ is composed of a cross-attention layer followed by feed-forward networks (FFN). For the input of $f$, we concatenate text tokens $x_q$ and visual tokens $x_v$, *i.e.*, $X = (x_q, x_v) \in \mathbb{R}^{N \times D}$, where $N$ is the number of input tokens and $D$ is the feature dimension. Then, the output feature $\tilde{X}$ can be calculated as:

$$\tilde{X} = f(X) \in \mathbb{R}^{N \times D}. \tag{5}$$

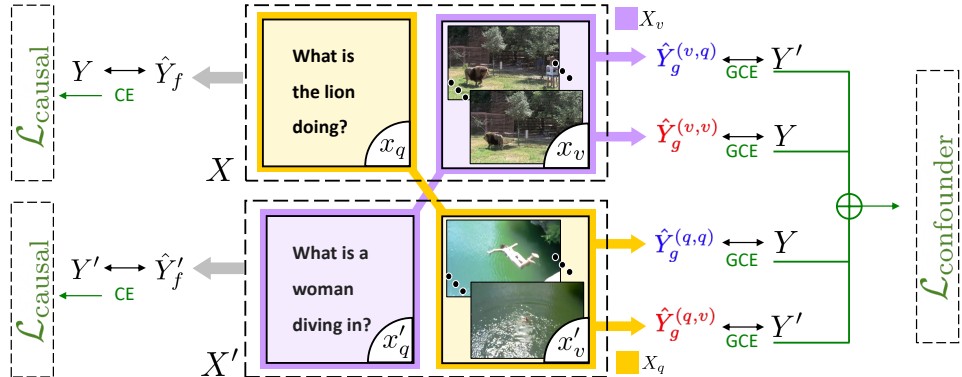

Figure 4: **Objective functions with unanswerable questions.** The first letter in the superscripts of the four outputs $\hat{Y}_g^{(*,*)}$ from $g$ stands for the modality of an input from the pair $X$, i.e., $\hat{Y}_g^{(q,*)}$ and $\hat{Y}_g^{(v,*)}$ denote outputs from $g$ given an input $X_q = (x_q, x'_v)$ and $X_v = (x'_q, x_v)$, respectively. The second letter denotes the modality an output is biased towards, which is also represented in color. $\hat{Y}^{(*,q)}$ and $\hat{Y}^{(*,v)}$ denotes that the output are biased towards text and video, respectively.

For unobserved confounders, we additionally introduce a set of learnable confounder queries $Z \in \mathbb{R}^{M \times D}$ as an input of $g$, where $M$ is the number of confounders and $D$ is the feature dimension. We also add two different modality encodings to inject information about each token's modality so that they could learn modality-specific bias. Concretely, a text-type encoding and a video-type encoding are added to individual confounder queries $Z[0 : M/2]$ and $Z[M/2 : M]$ The output of $f$, *i.e.*, $\tilde{X}$, is also used as an input of $g$ and cross-attended with $Z$. In detail, $Z$ is adopted as query, and $\tilde{X}$ is used as key and value of the encoder $g$. Then, the output $\tilde{Z}$ can be written as:

$$\tilde{Z} = \{\tilde{z} | \tilde{z} = g(\tilde{X}, z), \forall z \in Z\} \in \mathbb{R}^{M \times D}. \tag{6}$$

We additionally introduce two FFN $h_f$ and $h_g$ as prediction heads of $f$ and $g$, respectively. In short, the data encoder $f$ and $h_f$ perform the main VideoQA task, while the confounder encoder $g$ and $h_g$ learn and encode the bias in confounder queries $Z$, which will later be removed via causal intervention. The detailed objective functions to train each component are introduced in Sec. 3.3. Fig. 3 illustrates the overall architecture of our proposed framework.

### 3.3 TRAINING OBJECTIVE

**Debiased prediction.** Conventional approaches used $P(Y|X)$ in Eq. 3 as the output logit which is simply calculated with an additional FFN $h$ on the top of the encoder $f$, *i.e.*, $\hat{Y} = P(Y|X) = h(f(X))$. On the other hand, to remove the effect of confounders, we use a logit $P(Y|do(X))$ with causal intervention. Since $P(Y|do(X))$ is calculated with the Softmax function, it can be approximated by Normalized Weighted Geometric Mean (NWGM) (Xu et al., 2015) as follows:

$$\hat{Y}_f = P(Y|do(X)) = \sum_{z \in Z} P(Y|X, z)P(z) \approx P(Y|\sum_{z \in Z}(X + z)P(z)). \tag{7}$$

Then, the debiased output $\hat{Y}_f$ in Eq. 7 can be calculated with FFN $h_f$:

$$P(Y|\sum_{z \in Z}(X+z)P(z)) = h_f\left(\sum_{z \in Z}\left(\tilde{X} + z + g(\tilde{X}, z)\right)P(z)\right) = h_f\left(\sum_{z \in Z}\left(\tilde{X} + z + \tilde{z}\right)P(z)\right), \tag{8}$$

where $\tilde{X} = f(X)$ as defined in Eq. 5. Note that CLS token from $\tilde{X}$ is used when calculating $\tilde{X} + z + \tilde{z}$ in Eq. 8. In general, confounders are defined to be data-agnostic, *i.e.*, each sample in a dataset shares the same confounders. However, in Eq. 8, we consider not only *data-agnostic* confounders $Z$ but also *data-modulated* confounders $\tilde{Z}$. While debiasing the data-agnostic confounders leads to the model to alleviate dataset bias, by introducing additional data-modulated confounders $\tilde{Z}$, we also mitigate in-sample spurious correlations, *e.g.*, models tend to select the option which includes the visually salient 'object' in the video as an answer. We then apply the standard CE loss to perform the VideoQA task by using debiased logit with causal inference as:

$$\mathcal{L}_{causal} = \text{CE}(\hat{Y}_f, Y). \tag{9}$$

**Training confounders with unanswerable questions.** To alleviate the effect of confounders as in Eq. 7, it is important to let the confounder queries $Z \in \mathbb{R}^{M \times D}$ capture the bias during training. To achieve this, we first construct two unanswerable questions *i.e.*, $X_q = (x_q, x_v')$ and $X_v = (x_q', x_v)$ by pairing a text $x_q$ and video $x_v$ from a sample $X$ with label $Y$ with another video $x_v'$ and text $x_q'$ from a different sample $X'$ in mini-batch having label $Y'$. We then force the model to predict $Y$ or $Y'$ from these unanswerable questions. Since the model is unable to predict the proper answer given the unanswerable pair by relying only on the single modality but is forced to do so, the model inevitably learns text or video bias. Therefore, the model gets to only consider the spurious correlations to predict an answer. Fig. 4 shows the loss functions to train confounders with unanswerable questions.

In detail, two unanswerable pairs are forwarded to the encoder $f$ and $g$ with the confounders $Z$ as:

$$\tilde{X}_q, \ \tilde{X}_v = f(X_q), \ f(X_v)$$
$$\tilde{Z}_q, \ \tilde{Z}_v = \{\tilde{z} | \tilde{z} = g(\tilde{X}_q, z), \forall z \in Z\}, \ \{\tilde{z} | \tilde{z} = g(\tilde{X}_v, z), \forall z \in Z\}, \tag{10}$$

where $\tilde{Z}_q, \tilde{Z}_v \in \mathbb{R}^{M \times D}$. In Eq. 10, $\tilde{Z}_q$ and $\tilde{Z}_v$ indicate that output features of confounders $Z$, which are cross-attended with unanswerable pairs $X_q$ and $X_v$, respectively. As mentioned above, since the confounder is divided into two parts[2] to learn the text and video bias, we feed both $\tilde{Z}_q$ and $\tilde{Z}_v$, being separated into two parts respectively, to the FFN layer $h_g$ to output modality-biased predictions as.

$$\tilde{Z}_{q,q}, \ \tilde{Z}_{v,q}, \ \tilde{Z}_{q,v}, \ \tilde{Z}_{v,v} = \tilde{Z}_q[0:M/2], \ \tilde{Z}_v[0:M/2], \ \tilde{Z}_q[M/2:M], \ \tilde{Z}_v[M/2:M]$$
$$\hat{Y}_g^{(q,q)}, \ \hat{Y}_g^{(v,q)}, \ \hat{Y}_g^{(q,v)}, \ \hat{Y}_g^{(v,v)} = h_g(\tilde{Z}_{q,q}), \ h_g(\tilde{Z}_{v,q}), \ h_g(\tilde{Z}_{q,v}), \ h_g(\tilde{Z}_{v,v}), \tag{11}$$

Here, the former letter in the superscript of $\hat{Y}_g^{(*,*)}$ denotes the input modality which comes from the original pair $X$ when constructing the unanswerable pair (*e.g.*, An input question of $\hat{Y}_g^{(q,*)}$ is taken from $X_q$) and the latter denotes the modality an output would be biased towards. For instance, $\hat{Y}_g^{(v,q)}$, an output of $g$ given an input $X_v = (x_q', x_v)$, is desired to be text-biased, while $\hat{Y}_g^{(v,v)}$ is also an output of $g$ given an input $X_v = (x_q', x_v)$, but desired to be video-biased. In other words, $\tilde{Z}_{(*,q)}$ and $\hat{Y}_g^{(*,q)}$ denote text bias and a text-biased output from $g$. Similarly, $\tilde{Z}_{(*,v)}$ and $\hat{Y}_g^{(*,v)}$ denote video bias and a video-biased output from $g$.

Then, the loss function for training confounders to satisfy properties mentioned above is as follows:

$$\mathcal{L}_{\text{confounder}} = \text{GCE}(\hat{Y}_g^{(q,q)}, Y) + \text{GCE}(\hat{Y}_g^{(v,q)}, Y') + \text{GCE}(\hat{Y}_g^{(q,v)}, Y') + \text{GCE}(\hat{Y}_g^{(v,v)}, Y), \tag{12}$$

where GCE is the Generalized Cross Entropy (Zhang & Sabuncu, 2018) loss which will be further discussed below. For $\text{GCE}(\hat{Y}_g^{(q,q)}, Y)$ in Eq. 12, $\hat{Y}_g^{(q,q)}$ is a text-biased output desired to match $Y$ given an input $X_q = (x_q, x_v')$, *i.e.*, it is forced to learn the text bias regardless of the video, since the question $x_q$ and the answer $Y$ are from the same pair, while the irrelevant video $x_v'$ is from another sample in mini-batch. In the same way, $\text{GCE}(\hat{Y}_g^{(q,v)}, Y')$, where the input corresponding to $\hat{Y}_g^{(q,v)}$ is $(x_q, x_v')$, is encouraged to learn the video bias because $x_v'$ and $Y'$ are from the same pair. We can therefore train the confounders to learn both text bias and video bias with unanswerable questions in Eq. 12.

We can also amplify the bias and train the confounders to be more bias-toward by adopting GCE as:

$$\text{GCE}(p(x; \theta), y) = \frac{1 - p_y(x; \theta)^q}{q}, \tag{13}$$

where $p(x; \theta)$ is an output probability parameterized by $\theta$, $q \in (0, 1]$ is a smoothing parameter, and $y$ is a ground truth. The gradient of GCE loss is $p_y(x; \theta)^q$ times larger than the gradient of the standard CE loss, *i.e.*, $\frac{\partial \text{GCE}}{\partial \theta} = p_y(x; \theta)^q \cdot \frac{\partial \text{CE}}{\partial \theta}$. Therefore, GCE loss leads the model to be biased by placing larger weight on 'easier' samples with a high confidence score (Lee et al., 2021; Nam et al., 2020), inducing the model to be overfitted to easy shortcuts. Further details of GCE are in the supplement.

Then, the final loss function of our proposed algorithm is as follows:

$$\mathcal{L} = \mathcal{L}_{\text{causal}} + \mathcal{L}_{\text{confounder}}. \tag{14}$$

At the inference phase, we only use $\hat{Y}_f$, the output of causal intervention for the prediction.

---

[2]the first $M/2$ confounders $\tilde{Z}[0:M/2]$ denotes text confounders and the latter $M/2$ confounders $\tilde{Z}[M/2:M]$ means video confounders

Table 1: **Comparison on TGIF-QA, MSVD-QA, and MSRVTT-QA.** We report the accuracy for all datasets. TGIF-Action and TGIF-Transition are MCQA and TGIF-Frame, MSVD-QA, and MSRVTT-QA are OEQA.

| | TGIF-QA | | | MSVD-QA | MSRVTT-QA |
|---|---|---|---|---|---|
| **Method** | Action (MC) | Transition (MC) | Frame (OE) | OE | OE |
| ST-VQA (Jang et al., 2017) | 62.9 | 69.4 | 49.5 | - | - |
| Co-Mem (Gao et al., 2018) | 68.2 | 74.3 | 51.5 | 31.7 | 31.9 |
| HCRN (Le et al., 2020) | 75.0 | 81.4 | 55.9 | 36.1 | 35.6 |
| HGA (Jiang & Han, 2020) | 75.4 | 81.0 | 55.1 | 34.7 | 35.5 |
| QueST (Jiang et al., 2020) | 75.9 | 81.0 | 59.7 | 36.1 | 34.6 |
| Bridge2Answer (Park et al., 2021) | 75.9 | 82.6 | 57.5 | 37.2 | 36.9 |
| ClipBERT (Lei et al., 2021) | 82.9 | 87.5 | 59.4 | - | 37.4 |
| VIOLET (Fu et al., 2021) | 92.5 | 95.7 | **68.9** | 43.1 | - |
| MASN (Seo et al., 2021a) | 84.4 | 87.4 | 59.5 | 38.0 | 35.2 |
| QESAL (Liu et al., 2021) | 76.1 | 82.0 | 57.8 | 36.6 | 36.7 |
| CoMVT (Seo et al., 2021b) | - | - | - | 39.5 | 42.6 |
| IGV (Li et al., 2022) | - | - | - | 40.8 | 38.3 |
| HD-VILA (Xue et al., 2022) | 84.3 | 90.0 | 60.5 | - | 40.0 |
| HRNAT (Gao et al., 2022) | - | - | - | 38.2 | 35.3 |
| CASSG(Liu et al., 2022b) | 77.6 | 83.7 | 58.7 | 36.5 | 36.1 |
| HGQA (Xiao et al., 2022) | 76.9 | 85.6 | 61.3 | - | 38.6 |
| CMCIR(Liu et al., 2022a) | 78.1 | 82.4 | 62.3 | 43.7 | 38.9 |
| Mao et al. (2022) | 84.6 | 90.1 | 62.5 | - | 41.6 |
| **VoidQ** (Ours) | **93.5** | **97.3** | 67.0 | **46.2** | **43.2** |

**Prior probability $P(z)$.**    As for the prior probability $P(z)$, we introduce a learnable parameter $c \in \mathbb{R}^M$, *i.e.*, $P(z) = \text{Softmax}(c)$. However, since a large variance of $P(z)$ can make the training unstable, we apply exponential moving average (EMA) on $P(z)$ to stabilize the training. We also apply Dropout (Srivastava et al., 2014) on $P(z)$ and regularize a model from being overfitted on the particular confounders.

# 4 EXPERIMENTS

In this section, we evaluate the performance of VoidQ for both MCQA and OEQA settings on three benchmark VideoQA datasets: TGIF-QA, MSRVTT-QA, and MSVD-QA. In TGIF-QA, TGIF-Action and TGIF-Transition are conducted under the MCQA setting, predicting the proper answer among five options. For OEQA on TGIF-Frame, MSRVTT-QA, and MSVD-QA, we follow the conventional settings (Fu et al., 2021; Lei et al., 2021) to construct the answer candidates. In detail, the answer candidates of TGIF-Frame consist of 1,540 most frequent answers in the training set. Similarly, 1,500 and 1,000 most frequent answers in the training set are selected as the answer candidates for MSRVTT-QA, and MSVD-QA datasets. We also perform ablation studies to show that VoidQ is robust and generalizes well under distribution shifts. Our extensive qualitative analyses demonstrate that learnable confounder queries successfully capture the dataset bias, illustrated by our variant of Grad-CAM. Descriptions of datasets and implementation details are in the supplement.

## 4.1 QUANTITATIVE RESULTS

**TGIF-QA, MSVD-QA, and MSRVTT-QA.**    We compare VoidQ with previous VideoQA methods in Tab. 1. On TGIF-QA, VoidQ outperforms ClipBERT and VIOLET, which are foundation models pretrained on the large-scale dataset, especially by a margin of 1.0% on TGIF-Action, and 1.6% on TGIF-Frame compared to VIOLET. The performance of VoidQ also improves by 0.6% on MSRVTT-QA compared to CoMVT which is specifically designed for VideoQA tasks. On MSVD-QA, VoidQ obtains a 2.5% improvement over CMCIR which also conducts causal intervention.

**Ablation studies.**    Tab. 2 demonstrates the ablation studies in terms of the three components: confounder encoder $g$, unanswerable questions, and GCE loss. Without the unanswerable questions to train confounders, adding a confounder encoder $g$ degrades the performance from (a) 43.5% to (b) 41.9%. This result also evidences that the performance gain of VoidQ is not solely from increasing the model complexity. On the other hand, row (c) shows that the unanswerable questions significantly improve performance by a margin of 3.2% compared to (b). Adopting GCE loss (d) also leads to a further improvement in performance, since GCE loss helps the model better learn the confounder $Z$ by amplifying the dataset bias.

Table 2: **Ablation studies on MSVD.** '-' at $g$, the confounder encoder, denotes that we do not conduct the causal intervention and use the conventional likelihood $P(Y|X)$ for the prediction. UQ denotes the unanswerable questions with the standard CE loss. '✓' on both UQ and GCE stands for $\mathcal{L}_{\text{confounder}}$.

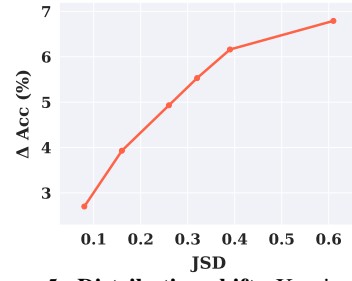

|     | $g$ | UQ | GCE | MSVD | MSRVTT→MSVD |
|-----|-----|-----|-----|------|-------------|
| (a) | -   | -   | -   | 43.5 | 36.3 |
| (b) | ✓   | -   | -   | 41.9 | - |
| (c) | ✓   | ✓   | -   | 45.1 | - |
| (d) | ✓   | ✓   | ✓   | **46.2** | **41.0** |

Figure 5: **Distribution shift.** X-axis: JSD between the train and modified test sets. Y-axis: Accuracy difference of the baseline and ours.

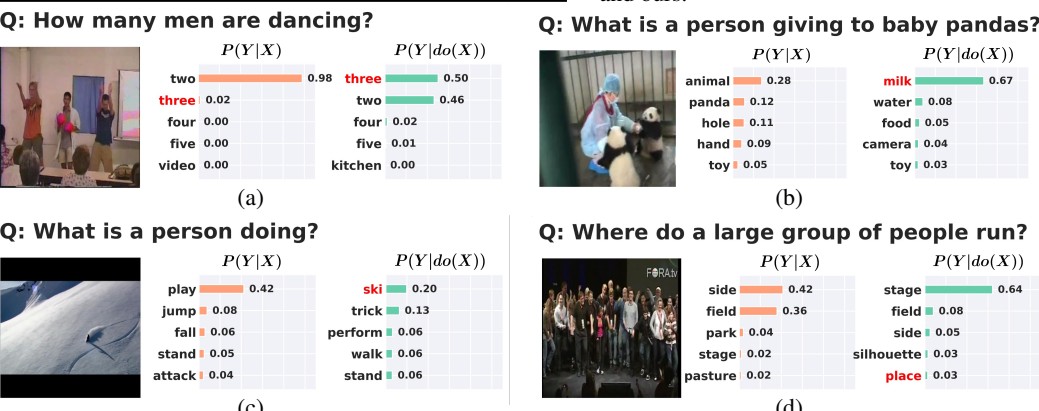

Figure 6: **Qualitative results on MSVD.** Confidence scores of the top-5 predicted answers using conventional likelihood $P(Y|X)$ and causal intervention $P(Y|do(X))$. Ground-truth answers are colored in red.

To show the generalizability of VoidQ, we also conduct experiments under the distribution/domain shift; we trained models on the MSRVTT training set and evaluated them on the MSVD test set (MSRVTT→MSVD). Jensen-Shannon Divergence, *i.e.*, $\text{JSD}(P,Q) = \frac{1}{2}\left(D_{\text{KL}}(P\|R) + D_{\text{KL}}(Q\|R)\right)$ where $R = \frac{1}{2}(P + Q)$, is adopted to quantify the label distribution distance between training and test sets. We observe that VoidQ provides a larger performance gain when the distribution shift increases. Where both training and evaluating on MSVD, JSD between the train and test set shows the relatively small value of $\text{JSD}(train_{\text{MSVD}}, test_{\text{MSVD}})$ 0.07. VoidQ obtains a 2.7% improvement from (a) 43.5% to (d) 46.2% in such setting. Whereas, the improvement increases to 4.7% when training on MSRVTT but evaluating on MSVD, where $\text{JSD}(train_{\text{MSRVTT}}, test_{\text{MSVD}}) = 0.26$.

We conduct additional experiments comparing model performances on the new test sets, including the standard test set, constructed by intentionally removing samples from the test set if the answer belongs in the top-1, 10, 20, 50, and 100 most frequent answer candidates. Such modification demonstrates how much a model is statistically biased. If a model is highly biased towards the dataset statistics, the model would perform worse when the frequent answers are removed. We compare the performance of VoidQ against the base VideoQA model without any debiasing scheme, which corresponds to (d) and (a) in Tab. 2, respectively. Fig. 5 illustrates the results of such experiments. The performance gap between the base model and the proposed model increases as the discrepancy between the train and test set enlarges. The performance gap of 2.7% between VoidQ and the baseline when JSD is 0.08 dramatically increases up to 6.79% when JSD is a larger value, 0.61. Two experiments done under distribution shift prove that VoidQ well alleviates the statistical bias, helping the model to successfully perform on the dataset that differs from the train set.

## 4.2 QUALITATIVE ANALYSES

**How is the model debiased?** Four examples in Fig. 6 illustrate how models' predictions are corrected via causal intervention $P(Y|do(X))$. Without the causal intervention, *i.e*, $P(Y|X) = h_f(f(X))$, the model is prone to answer questions based on a single modality. For instance, in Fig. 6a, $P(Y|X)$ predicts 'two' as the answer since about 84% answers to the 'How many' questions in the training set are 'two' without considering an input video. Another example in Fig. 6b $P(Y|X)$ predicts 'animal' or 'panda' as an answer, focusing on visually salient objects, overlook-

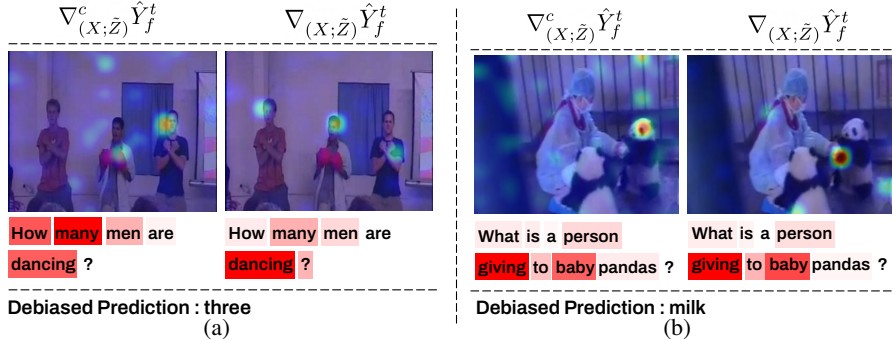

Figure 7: **GradCAM on MSVD.** We visualize $\nabla^c_{(X;\tilde{Z})}\hat{Y}^t_f$ and $\nabla_{(X;\tilde{Z})}\hat{Y}^t_f$ on two samples to show where the confounders look at in the input.

ing the input question. However, after conducting the causal intervention $P(Y|do(X))$, output are corrected to 'three' and 'milk' by thoroughly considering the previously neglected video and text input, respectively. VoidQ also succeeds predicting the answer 'ski' in Fig. 6c, although the answer 'play' appeared 51 times more than 'ski' during training. Finally, Fig. 6d shows that VoidQ plausibly predicts 'stage', overcoming the dataset bias that 'side' and 'field' are the two most frequent answers to the 'Where' questions.

**Where do the confounders look at?** We investigate which parts in a video or which words in a question are taken into account by confounder queries to debias the predictions of a VideoQA model. To consider the gradient flows through $\tilde{Z}$, we modify GradCAM and Counterfactual Grad-CAM (Selvaraju et al., 2017) denoted as $\nabla_{(X;\tilde{Z})}\hat{Y}^t_f$ and $\nabla^c_{(X;\tilde{Z})}\hat{Y}^t_f$, respectively. They are computed as:

$$\nabla_{(X;\tilde{Z})}\hat{Y}^t_f := \mathrm{ReLU}\left(\sum_{\tilde{z}\in\tilde{Z}} \frac{\partial \hat{Y}^t_f}{\partial \tilde{z}} \cdot \frac{\partial \tilde{z}}{\partial X}\right), \qquad \nabla^c_{(X;\tilde{Z})}\hat{Y}^t_f := \mathrm{ReLU}\left(\sum_{\tilde{z}\in\tilde{Z}} -\frac{\partial \hat{Y}^t_f}{\partial \tilde{z}} \cdot \frac{\partial \tilde{z}}{\partial X}\right), \quad (15)$$

where $t$ is the target label in question. $\nabla_{(X;\tilde{Z})}\hat{Y}^t_f$ illustrates where confounder $\tilde{Z}$ focused on to bolster correct predictions. Conversely, $\nabla^c_{(X;\tilde{Z})}\hat{Y}^t_f$ reveals where confounder $\tilde{Z}$ focused on to suppress falsely correlated cues that cause bias. Fig. 7 illustrates $\nabla_{(X;\tilde{Z})}\hat{Y}^t_f$ and $\nabla^c_{(X;\tilde{Z})}\hat{Y}^t_f$ on two examples shown in Fig. 6a and 6b which contains text and video bias, respectively. Fig. 7a (left) reveals that "How many" is strongly highlighted by $\nabla^c_{(X;\tilde{Z})}\hat{Y}^t_f$, implying the phrase negatively influenced to predict a correct answer. Interestingly, it matches our assumption that "How many" can be considered as the confounder leading a model to predict 'two' as the answer. On the other hand, $\nabla_{(X;\tilde{Z})}\hat{Y}^t_f$ highlights "dancing" most, meaning that VoidQ focuses on the right context to output the number of 'dancing men'. Similarly in Fig. 7b including video bias, $\nabla^c_{(X;\tilde{Z})}\hat{Y}^t_f$ and $\nabla_{(X;\tilde{Z})}\hat{Y}^t_f$ focus on the 'panda' and 'milk' respectively, which matches our notion that looking at 'panda' in the video hinders the model to predict correct answer looking at 'milk' in the video helps debiasing.

## 5 CONCLUSION

In this work, we propose a novel debiasing framework for VideoQA, dubbed VoidQ, which trains confounder queries by answering unanswerable questions and utilizes the trained confounders to remove the dataset bias via causal intervention. Concretely, we adopt causal intervention to cut-off the relation between confounders $Z$ and input $X$ so that the model predicts the correct answer $Y$ with bias removed. Since the causal intervention is not applicable when confounders are unobserved, we additionally introduce a training scheme that leverages *unanswerable* questions to let learnable confounder queries capture the dataset bias. We demonstrate the effectiveness of our method by validating the proposed architecture on various benchmark datasets, and provide qualitative analyses showing that confounders are well learned to capture the dataset bias and properly removed.

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

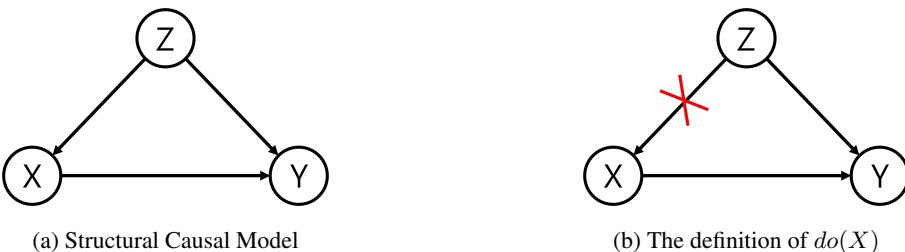

(a) Structural Causal Model          (b) The definition of $do(X)$

Figure 8: **Illustration of Structural Causal Model (SCM) and *do*-calculus definition**.

# A  APPENDIX

# B  ADDITIONAL PRELIMINARIES

## B.1  CAUSAL INFERENCE

**Structural Causal Model (SCM).**  SCM is a statistical model representing the causal relationship between variables in the graph structure Glymour et al. (2016). In the causal graph, each variable is denoted by nodes, and 'causation' between two different variables is denoted by a directed edge between nodes. Fig. 8a illustrates an example of SCM representation in graph form. The edge $X \rightarrow Y$ in the graph implies that $X$ is the 'cause' of $Y$. Also, $Z$ in the graph represents a 'confounder', which simultaneously affects both $X$ and $Y$, therefore making it difficult to find a true effect of $X$ on $Y$. Such confounder induces the spurious correlation between $X$ and $Y$ through the backdoor path between $X$ and $Y$. A backdoor path is formally defined as any path from $X$ to $Y$ that starts with an arrow pointing to $X$ (Yang et al., 2021), such as $X \leftarrow Z \rightarrow Y$ in 8a. To find out the true causal relationship between $X$ and $Y$, the causal intervention with *do*-calculus $P(Y|do(X))$ is applied to cut-off the relationship $Z \rightarrow X$, as illustrated in 8b, therefore removing the spurious correlation induced by $Z$. Backdoor adjustment is a widely adopted approach to deconfound the effect of the confounders $Z$ using the *do*-calculus, which we further concretize in the very following section.

**The backdoor adjustment.**  Given a directed acyclic graph consisting of $X$, $Y$, and $Z$ as in 8a, backdoor adjustment can be applied to reveal the true causal effect of the $X$ on $Y$ given the confounder $Z$. By Bayes' theorem, $P(Y|X)$ can be expressed as follows:

$$P(Y|X) = \sum_{z \in Z} P(Y|X, Z = z)P(Z = z|X). \tag{16}$$

The causal intervention with *do*-calculus $P(Y|do(X))$ mentioned in the previous section is then formally defined as below:

$$P(Y|do(X)) = \sum_{z \in Z} P(Y|X, Z = z)P(Z = z). \tag{17}$$

Through the backdoor adjustment, the true causal relationship between $X$ and $Y$, which is denoted as $P(Y|do(X))$ is measured without any effect of the confounder $Z$.

**Normalized Weighted Geometric Mean (NWGM).**  To approximate $P(Y|do(X))$, we use NWGM. Before dealing with NWGM, we first revisit the definition of Weighted Geometric Mean (WGM). Given a discrete variable X and its distribution $P(X)$, the expectation of $f(x)$ is defined as:

$$\mathbb{E}_x[f(x)] = \sum_{x \in X} f(x)P(x). \tag{18}$$

The Weighted Geometric Mean (WGM), an approximation of $\mathbb{E}_x[f(x)]$ is defined as follows:

$$\text{WGM}(f(x)) = \prod_{x \in X} f(x)^{P(x)}. \tag{19}$$

If the activation function of $f(x)$ is a composition of a function $g(x)$ followed by an exponential function, *i.e.*, $f(x) = \exp(g(x))$, Eq. 19 can be reformulated as:

$$
\begin{aligned}
\text{WGM}(f(x)) &= \prod_{x \in X} \exp[g(x)]^{P(x)} = \prod_{x \in X} \exp[g(x)P(x)] \\
&= \exp(\sum_{x \in X} g(x)P(x)) = \exp\{\mathbb{E}_x[g(x)]\}.
\end{aligned}
\tag{20}
$$

Interpreting WGM in the perspective of deep learning, $f(x)$ can be regarded as a neural network whose last activation function is the softmax function. Therefore, Xu et al. (2015) and Yang et al. (2021) approximate the expectation of the $f(x)$ using the WGM as follows:

$$
\mathbb{E}_x[f(x)] \approx \text{WGM}(f(x)) = \exp\{\mathbb{E}_x[g(x)]\}
\tag{21}
$$

To guarantee that output logits can be interpreted as a probability, NWGM, a normalized version of WGM, is applied so that the sum of output logits adds up to one, and it is formally defined as:

$$
\begin{aligned}
\text{NWGM}(f(x)) &= \frac{\prod_x \exp(g(x))^{P(x)}}{\sum_j \prod_x \exp(g(x))^{P(x)}} \\
&= \frac{\exp(\mathbb{E}_x[f(x)])}{\sum_j \exp(\mathbb{E}_x[f(x)])} \\
&= \text{Softmax}(\mathbb{E}_x[f(x)])
\end{aligned}
\tag{22}
$$

Adopting the WGM defined above to our model, $P(Y|do(X))$ can be approximated as below, where $P(Y|X, z) = Softmax(g(X, z)) \propto \exp(g(X, z))$:

$$
\begin{aligned}
P(Y|do(X)) &= \mathbb{E}_z[P(Y|X, z)] \\
&= \mathbb{E}_z[\exp(g(X, z))] \\
&\approx \exp(\mathbb{E}_z[g(X, z)]) \\
&= \exp\{\sum_{z \in Z} (f(X) + z + \tilde{z}) P(z)\}.
\end{aligned}
\tag{23}
$$

where $g(X, z) = f(X) + z + \tilde{z}$. Then, we apply NWGM to normalize Eq. 23 as to get final deconfounded prediction probabilities $P(Y|do(X))$ as follows:

$$
\begin{aligned}
P(Y|do(X)) &\approx \text{Softmax}(\mathbb{E}_z[g(X, z)]) \\
&= \text{Softmax}\{\sum_{z \in Z} (f(X) + z + \tilde{z}) P(z)\}.
\end{aligned}
\tag{24}
$$

## B.2 GENERALIZED CROSS ENTROPY (GCE) LOSS

**GCE.** GCE loss was first proposed as a generalized loss taking advantage of both Mean Absolute Error (MAE) loss, and Categorical Cross Entropy (CCE) loss by Zhang & Sabuncu (2018). Given an input $x$, the ground truth one-hot vector $y$, and the set of parameters $\theta$ of the classifier $f$, MAE and CCE loss are formally defined as below in the common case where the softmax is followed by the classification layer:

$$
\begin{aligned}
\mathcal{L}_{MAE}(f(x; \theta), y) &= ||y - f(x; \theta)||_1 \\
\mathcal{L}_{CCE}(f(x; \theta), y) &= -\sum_{j=1}^{C} y_j \log f_j(x; \theta),
\end{aligned}
\tag{25}
$$

where $C$ denotes the number of target classes, $y_j$ and $f_j$ denote the $j$-th element of $y$ and the $j$-th prediction of $f$. The gradient of loss functions with respect to parameter $\theta$ is as follows:

$$
\begin{aligned}
\frac{\partial \mathcal{L}_{MAE}(f(x; \theta), y)}{\partial \theta} &= -\nabla_\theta f_y(x; \theta) \\
\frac{\partial \mathcal{L}_{CCE}(f(x; \theta), y)}{\partial \theta} &= -\frac{1}{f_y(x; \theta)} \nabla_\theta f_y(x; \theta),
\end{aligned}
\tag{26}
$$

where $f_y$ denotes the element of the output logit corresponding to the ground-truth label. As formulated in Eq. 26, CCE emphasizes samples with larger $1/f_y(x; \theta)$, or smaller $f_y(x; \theta)$. On the contrary, MAE equally treats every sample with the same weight. The fact that MAE does not place a larger weight on difficult samples makes MAE robust to noisy labels, but it also makes training difficult since every sample is treated equally so that challenging examples are not learned enough. In contrast, optimizing a model using CCE is easier due to larger weights being given to challenging samples. However, CCE is sensitive to noisy labels, since the model could easily be overfitted to such noisy samples which are intrinsically difficult due to label noise. Then GCE loss can be viewed as a generalization between MAE and CCE loss, and is formally defined as below:

$$\mathcal{L}_{GCE}(f(x; \theta), y) = \frac{1 - p_y(x; \theta)^q}{q}, \tag{27}$$

where $q \in (0, 1]$ is a smoothing parameter. The gradient of $\mathcal{L}_{GCE}$ with respect to $\theta$ is as follows:

$$\frac{\partial \mathcal{L}_{GCE}(f(x; \theta), y)}{\partial \theta} = f_y(x; \theta)^q (-\frac{1}{f_y(x; \theta)} \nabla_\theta f_y(x; \theta)) = f_y(x; \theta)^q \frac{\partial \mathcal{L}_{CCE}}{\partial \theta}$$
$$= -f_y(x; \theta)^{q-1} \nabla_\theta f_y(x; \theta) = f_y(x; \theta)^{q-1} \frac{\partial \mathcal{L}_{MAE}}{\partial \theta}. \tag{28}$$

Therefore, $\mathcal{L}_{GCE}$ additionally weights each sample by $f_y(x; \theta)^q$ times compared to CCE loss, weighting difficult samples less. Also, it weights each sample by $f_y(x; \theta)^{q-1}$ times compared to MAE loss, giving larger weight to difficult examples compared to MAE loss. If $q$ is properly chosen, GCE can therefore act as a generalized loss that is more robust than CCE and easier to train than MAE, achieving a balanced trade-off between two losses.

**GCE Loss in Computer Vision.** By the fact that GCE loss gives smaller weights to 'difficult' examples compared to conventional CCE loss, Lee et al. (2021) and Nam et al. (2020) propose capturing bias in the model by leveraging GCE loss to train a 'biased network', which is overfitted to easy samples, which corresponds to 'bias' or 'spurious correlation' existing in the dataset. Both works train the model with GCE loss to achieve the model to be biased by focusing on the "easier" samples compared to the conventional CCE.

## B.3 GRADCAM

The standard GradCAM (Selvaraju et al., 2017) of prediction $\hat{Y}_f^t$ with respect to input $X$ can be calculated as:

$$\nabla_X \hat{Y}_f^t := \text{ReLU}\left(\frac{\partial \hat{Y}_f^t}{\partial X}\right) = \text{ReLU}\left(\frac{\partial \hat{Y}_f^t}{\partial \tilde{X}} \cdot \frac{\partial \tilde{X}}{\partial X} + \frac{\partial \hat{Y}_f^t}{\partial \tilde{Z}} \cdot \frac{\partial \tilde{Z}}{\partial X}\right), \tag{29}$$

since the information of input $X$ is divided into two streams, *i.e.*, $\tilde{X}$ and $\tilde{Z}$, and merged to make the prediction $\hat{Y}_f^t$. Here, $t$ is the target label in question so visualization of Eq. 29 illustrates which parts in the input affect predicting the label $t$. However, Eq. 29 takes into account the gradient flows through both $\tilde{X}$ and $\tilde{Z}$ although we want to know only the flows through confounders $\tilde{Z}$ to visualize where the confounders look at. So we define and visualize the gradient through $\tilde{Z}$ as:

$$\nabla_{(X; \tilde{Z})} \hat{Y}_f^t := \text{ReLU}\left(\frac{\partial \hat{Y}_f^t}{\partial \tilde{Z}} \cdot \frac{\partial \tilde{Z}}{\partial X}\right) = \text{ReLU}\left(\sum_{\tilde{z} \in \tilde{Z}} \frac{\partial \hat{Y}_f^t}{\partial \tilde{z}} \cdot \frac{\partial \tilde{z}}{\partial X}\right), \tag{30}$$

which is consistent with Eq. 15 of the main paper. Here, $\frac{\partial \hat{Y}_f^t}{\partial \tilde{Z}}$ means *how much $\tilde{Z}$ affects the prediction $\hat{Y}_f^t$* and $\frac{\partial \tilde{Z}}{\partial X}$ means *what the confounder $\tilde{Z}$ looks at in the input $X$*, so $\nabla_{(X; \tilde{Z})} \hat{Y}_f^t$ indicates what the confounder $\tilde{Z}$, affecting the prediction $\hat{Y}_f^t$, looks at in the input $X$. Compared to $\nabla_{(X; \tilde{Z})} \hat{Y}_f^t$, by simply adding a negative sign, Counterfactual GradCAM $\nabla_{(X; \tilde{Z})}^c \hat{Y}_f^t$ is defined as:

$$\nabla_{(X; \tilde{Z})}^c \hat{Y}_f^t := \text{ReLU}\left(\sum_{\tilde{z} \in \tilde{Z}} -\frac{\partial \hat{Y}_f^t}{\partial \tilde{z}} \cdot \frac{\partial \tilde{z}}{\partial X}\right), \tag{31}$$

meaning that where confounder $\tilde{Z}$ focuses on to suppress the prediction $\hat{Y}_f^t$.

---

**Algorithm 1** Overall Algorithm

---

**Inputs:** sample $\{X = (x_q, x_v), Y\}$, negative sample $\{X' = (x'_q, x'_v), Y'\}$, confounder queries $Z$, number of confounder queries $M$

**Parameters:** prior probability $c$, data encoder $f$, confounder encoder $g$, FFN $\{h_f, h_g\}$

1: $X_q, \ X_v \leftarrow (x_q, x'_v), \ (x'_q, x_v)$
2: $\tilde{X}, \ \tilde{X}_q, \ \tilde{X}_v \leftarrow f(X), \ f(X_q), \ f(X_v)$
3: $\tilde{Z}, \tilde{Z}_q, \tilde{Z}_v \leftarrow \{\tilde{z}|\tilde{z} = g(\tilde{X}, z), \forall z \in Z\}, \{\tilde{z}|\tilde{z} = g(\tilde{X}_q, z), \forall z \in Z\}, \{\tilde{z}|\tilde{z} = g(\tilde{X}_v, z), \forall z \in Z\}$
4: $\hat{Y}_f \leftarrow h_f \left( \sum_{z \in Z} \left( \tilde{X} + z + \tilde{z} \right) c_z \right)$            $\triangleright c_z$ is a prior probability of $z$
5: $\tilde{Z}_{q,q}, \ \tilde{Z}_{q,v}, \ \tilde{Z}_{v,q}, \ \tilde{Z}_{v,v} \leftarrow \tilde{Z}_q[0 : M/2], \ \tilde{Z}_q[M/2 : M], \ \tilde{Z}_v[0 : M/2], \ \tilde{Z}_v[M/2 : M]$
6: $\hat{Y}_g^{(q,q)}, \ \hat{Y}_g^{(q,v)}, \ \hat{Y}_g^{(v,q)}, \ \hat{Y}_g^{(v,v)} \leftarrow h_g(\tilde{Z}_{q,q}), \ h_g(\tilde{Z}_{q,v}), \ h_g(\tilde{Z}_{v,q}), \ h_g(\tilde{Z}_{v,v})$
7: $\mathcal{L}_{\text{causal}} \leftarrow \text{CE}(\hat{Y}_f, Y)$
8: $\mathcal{L}_{\text{confounder}} \leftarrow \text{GCE}(\hat{Y}_g^{(q,q)}, Y) + \text{GCE}(\hat{Y}_g^{(q,v)}, Y') + \text{GCE}(\hat{Y}_g^{(v,q)}, Y') + \text{GCE}(\hat{Y}_g^{(v,v)}, Y)$
9: $\mathcal{L} \leftarrow \mathcal{L}_{\text{causal}} + \mathcal{L}_{\text{confounder}}$
10: **return** $\mathcal{L}$

---

## C    EXPERIMENTAL SETTINGS

### C.1    DATASET

We validate the proposed model on four benchmark datasets: **TGIF-QA** (Li et al., 2016; Jang et al., 2017), **MSVD-QA** (Chen & Dolan, 2011; Xu et al., 2017), and **MSRVTT-QA** (Xu et al., 2016; 2017). **TGIF-QA** consists of 103,913 QA pairs from 56,720 GIFs and includes three multiple-choice VideoQA tasks: repetition count, repeating action, and state transition, along with an open-ended frameQA task reasoning on a single frame. **MSVD-QA** and **MSRVTT-QA** are both open-ended VideoQA datasets with descriptive QA tasks, while **MSRVTT-QA** consists of more complex and longer 10,000 trimmed videos and larger 243,000 QA pairs compared to **MSVD-QA** with 1,970 trimmed videos and 50,500 QA pairs.

### C.2    IMPLEMENTATION DETAILS.

**Model architecture.**    We adopt the Transformer (Vaswani et al., 2017) architecture with 12 layers for both the data encoder $f$ and the confounder encoder $g$. Concretely, for the data encoder $f$, visual tokens $X_v$ and text tokens $X_q$ are concatenated with an additional [CLS] token to form an input $X = (x_q, x_v) \in \mathbb{R}^{N \times D}$. To build $x_v$, we sample 3 frames per single input video. Each frame has a spatial resolution of 224×224, and is patchified into 14×14 patches with the size of 16×16 for each. For text token $x_q$, we set 40 as the max length of the input text sequence. An input text is then tokenized to have a hidden dimension of $D = 768$. After concatenating $x_q$ and $x_v$, modality encoding is added to input tokens having corresponding modalities. When conducting cross-attention in $g$, we apply a stop-gradient operation to $\tilde{X}$ so that it could not be affected by $\mathcal{L}_{\text{confounder}}$. Also, we use $M = 128$ for the number of confounder query tokens.

**Training details.**    For training, the initial learning rate is set to $10^{-4}$ with cosine decay and warmup applied until 10% of the total training step is done. We train the models with AdamW (Loshchilov & Hutter, 2017) optimizer with a weight decay rate of 0.01. The probability of confounder dropout is 0.15. Our backbone encoders are pretrained on Webvid (Bain et al., 2021), YT-Temporal 180M (Zellers et al., 2021), HowTo100M (Miech et al., 2019), CC3M (Sharma et al., 2018), CC12M (Changpinyo et al., 2021), COCO (Lin et al., 2014), VisualGenome (Krishna et al., 2017), and SBU (Ordonez et al., 2011) as in Fu et al. (2021) and Wang et al. (2022). All the experiments are conducted on 4 × Tesla A100 GPUs.

**MCQA details.**    We concatenate each option and the question and insert the [SEP] token between them to construct the text token sequence. To efficiently calculate $\mathcal{L}_{\text{confounder}}$, we only take into account two negative pairs $(X_q, Y)$ and $(X_v, Y')$ instead of four negative pairs including $(X_q, Y')$ and

**Q: What does the man do 4 times ?**

(a) TGIF-Action

**Q: What does the man do after lift bucket ?**

(b) TGIF-Transition

**Q: What is the small kitten licking while laying in the covers ?**

(c) TGIF-Frame

Figure 9: **Qualitative results on TGIF.** Confidence scores of the top-5 predicted answers using conventional likelihood $P(Y|X)$ and causal intervention $P(Y|do(X))$. Ground-truth answers are colored in red.

$(X_v, Y)$, *i.e.*, $\mathcal{L}_{\text{confounder}} = \text{GCE}(\hat{Y}_g^{(q,q)}, Y) + \text{GCE}(\hat{Y}_g^{(v,q)}, Y')$. This is because it is cumbersome to forward all the combinations of negative pairs including concatenated text token sequences for each option.

# D    OVERALL ALGORITHM

The overall algorithm to train our proposed framework is formulated in Alg. 1.

# E    FURTHER QUALITATIVE ANALYSES

## E.1    DEBIASED PREDICTION

Fig. 9 illustrates how models' predictions are corrected via causal intervention $P(Y|do(X))$. We discuss the detected biases for three representative question types in TGIF.

**TGIF-Action.** As shown in Fig. 9a, the model tends to predict 'shake head' without considering the visual context before the causal intervention. On the other hand, the prediction is corrected to 'rub something with fingers' after the causal intervention. We believe that this case is biased to text since the 'shake head' co-occurs 141 times more than 'rub' with the word 'man' in the question. Here, the word 'man' serves as the confounder inducing the text bias.

**TGIF-Transition.** Fig. 9b shows the text bias. In this case, the 'smile' & 'man' pair co-occurs 169 times more than 'dump' & 'man' pair, which leads the model to predict 'smile' only considering the text. However, after the causal intervention with $P(Y|do(X))$, the model predicts the answer correctly.

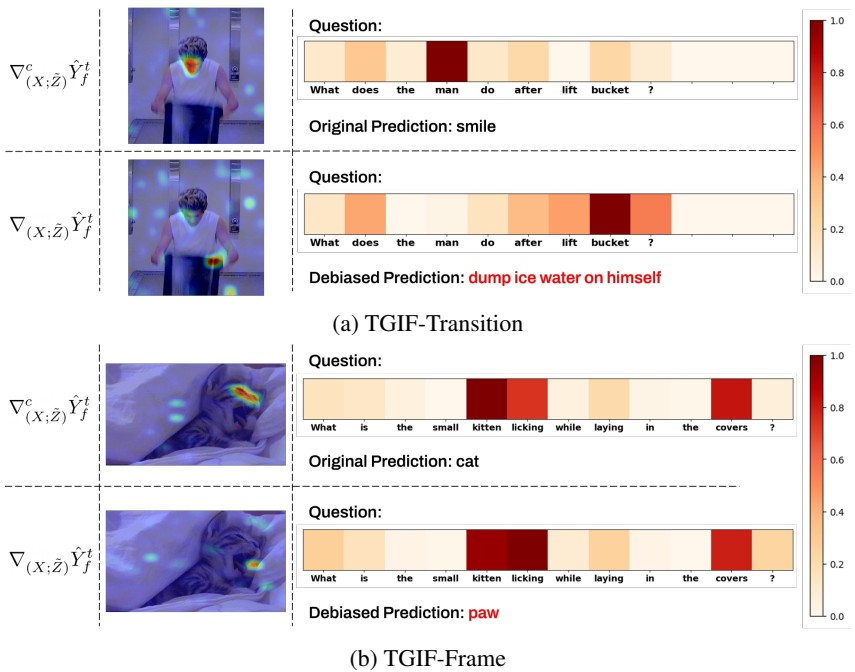

(a) TGIF-Transition

(b) TGIF-Frame

Figure 10: **GradCAM on TGIF.** We visualize $\nabla^c_{(X;\tilde{Z})}\hat{Y}^t_f$ and $\nabla_{(X;\tilde{Z})}\hat{Y}^t_f$ on two samples to show where the confounders look at in the input.

**TGIF-Frame.** Fig. 9b illustrates the video-biased case. $P(Y|X)$ is likely to focus on the visually salient object 'cat' without considering the question. By applying causal intervention, the video bias is alleviated and the prediction is corrected to 'paw' from 'cat' considering both the video and question.

### E.2 GRADCAM VIUSALIZATION

In the main paper, using variants of GradCAM, we have investigated which words in a question or which parts in a video are taken into account by confounder queries to debias the predictions of VideoQA model with GradCAM $\nabla_{(X;\tilde{Z})}\hat{Y}^t_f$ and Counterfactual GradCAM $\nabla^c_{(X;\tilde{Z})}\hat{Y}^t_f$ for MSVD. For TGIF-QA, Fig. 10a shows the same QA pair with Fig.e 9b, which is biased to the text. The word 'man' is strongly highlighted by counterfactual GradCAM $\nabla^c_{(X;\tilde{Z})}\hat{Y}^t_f$ implying it negatively influences to predict the correct answer. On the other hand, GradCAM $\nabla_{(X;\tilde{Z})}\hat{Y}^t_f$ focuses on the word 'bucket' to output a correct answer 'dump ice water on himself'. This is consistent with our observation that the word 'man' is considered as the text confounder hindering the model from predicting correctly. In Fig. 10b, the video-biased QA pair come from Fig. 9c, Counterfactual GradCAM $\nabla^c_{(X;\tilde{Z})}\hat{Y}^t_f$ shows that the object 'cat/kitten' which is visually salient object in the video hinders the model to predict the proper answer. However, GradCAM $\nabla_{(X;\tilde{Z})}\hat{Y}^t_f$ focuses on the object 'paw' in the video so the model correctly predicts the answer. This indicates that the object 'cat' serves as the video confounder in the video-biased sample.

## F FURTHER ABLATION STUDIES

### F.1 CONFOUNDER QUERIES $Z$

As shown in Tab. 3, the performance slightly decreases by 0.8% when only using the text confounder queries. On the other hand, the performance decreases by 1.9% when only using the video confounder queries. This indicates that the dataset has a stronger text bias than video.

Table 3: **Ablation study on the type of confounders.** $Z[0:M]$, $Z[0:M/2]$, and $Z[M/2:M]$ refer to entire confounder queries, text confounder queries, and video confounder queries, respectively.

| | $Z[0:M]$ | $Z[0:M/2]$ | $Z[M/2:M]$ |
|---|---|---|---|
| MSVD | 46.4 | 45.2 | 44.5 |

### F.2 NUMBER OF CONFOUNDER QUERIES

As shown in the Tab. 4, the model performs best when the number of confounders is near 64 or 128. In our experiments, we used 128 confounder queries.

Table 4: **Ablation study on the number of confounder queries.**

| | 4 | 8 | 16 | 32 | 64 | 128 | 256 | 512 | 1024 |
|---|---|---|---|---|---|---|---|---|---|
| MSVD | 44.8 | 45.0 | 45.7 | 46.3 | **46.4** | **46.4** | 46.2 | 46.1 | 45.4 |

