# OpenReview forum: "Answer Me if You Can: Debiasing Video Question Answering via Answering Unanswerable Questions"
_ICLR.cc/2023/Conference — Submitted to ICLR 2023_

### Official Review · Reviewer_H5yq · 2022-10-20

**Confidence:** 4
**Correctness:** 3
**Technical Novelty And Significance:** 3
**Empirical Novelty And Significance:** 2
**Recommendation:** 6

**Clarity, Quality, Novelty And Reproducibility:**

The paper is clear. For reproducibility may need the authors to open-source the code.

**Strength And Weaknesses:**

Strength
- Casual intervention is gaining popularity as a way to debias vision-and-language models that leverage statistical correlations to make predictions. An important limitation with casual intervention is that the so-called confounders need to be known in advance. Previous work approximates those condounders with, for example, the list of objects in COCO datasets. In contrast, this paper proposes to learn confounders by using unmatching question-video pairs and forcing the model to make predictions based on their statistical correlations.
- The experiments show that the method achieves high accuracy. Ablation studies shows that all the parts of the proposed framework are relevant.

Weaknesses
- A question that arises and it is not very clear in the paper is how to ensure that the unmatching question-video pairs are really unrelated. Just choosing random pairs does not ensure that the video and the question are irrelevant. For example, a "how many people" question can still be relevant in a different video.

**Summary Of The Paper:**

The paper proposes a framework for debiasing video question answering (VideoQA) with casual intervention. The novelty relies in that the confounders do not need to be know in advance. Instead, the framework uses pairs of "unanswerable" questions at training time to find them. Experiments are conducted on three standard VideoQA datasets, achieving better accuracy than previous methods on all of them. Ablation studies and quantitative results are also provided.

**Summary Of The Review:**

The paper proposes an interesting solution to the problem of causal intervention in VideoQA without knowing confounders in advance, which is a more realistic setting than using a list of objects as confounders. Experiments are technically sound and show better performance than previous work.

---

> ### Author Response · Authors · 2022-11-14
> **Response for Reviewer H5yq**
>
> We appreciate **Reviewer H5yq** for the constructive comments and support for our motivation and solution to the problem. We will address your concerns, hoping for more vigorous support for our paper.
>
> ---
>
> **Concern 1:**
>
> How to ensure that the unmatching question-video pairs are really unrelated.
>
> **Answer:**
>
> This question is closely related to the Reviewer aBdW’s Concern 3. For the convenience of the reviewer, we here write the answer again. We here provide additional analysis to check whether newly-created pairs are unanswerable or not. We randomly sample 100 instances from TGIF-Action (MCQA) and MSRVTT (OEQA) respectively to show the statistics of unanswerable QA pairs. For each QA pair, we manually classify it into three categories: (A) Unanswerable QA pair, (B) Wrong answerable QA pair (answerable but inducing the wrong answer), and (C) answerable QA pair. The statistics of each dataset are as follows:
> |  | TGIF-Action (MCQA) | MSRVTT (OEQA) |
> | --- | :---: | :---: |
> | (A) Unanswerable | 86% | 80% |
> | (B) Wrong answerable | 7% | 15% |
> | (C) Answerable | 7% | 5% |
>
> Since the model can also learn the bias when it is forced to predict the wrong answer, we consider case (B) as (A), either. Then, the proportions of answerable question/answer pairs are only 7% and 5% in MCQA and OEQA, respectively. We believe that the portion of answerable question/answer pairs is sufficiently small and the bias by GCE loss ensures that our confounder queries learn biases as in the literature of [1, 2].
>
> [1] Nam et al., “Learning from Failure: Training Debiased Classifier from Biased Classifier”, NeurIPS 2020.
>
> [2] Lee et al., “Learning Debiased Representation via Disentangled Feature Augmentation”, NeurIPS 2021 (Oral).

---

> > ### Author Response · Authors · 2022-12-08
> > **A gentle reminder for Reviewer H5yq**
> >
> > Dear **Reviewer H5yq**, thank you again for your effort in reviewing our paper.
> >
> > The author-reviewer discussion will be closed soon.
> > Through rebuttal, we have addressed all your concerns, and we believe that our responses have answered your suggestions and questions.
> > Please let me know if you have further questions or concerns.
> > We look forward to your feedback and discussion.
> >
> > Sincerely, Authors

---

### Official Review · Reviewer_EnYB · 2022-10-23

**Confidence:** 4
**Correctness:** 4
**Technical Novelty And Significance:** 3
**Empirical Novelty And Significance:** 3
**Recommendation:** 6

**Clarity, Quality, Novelty And Reproducibility:**

The paper conduct a good original research on debasing video QA. It is relatively easy to reproduce the model architecture based on the descriptions in the paper.

**Strength And Weaknesses:**

Strengths:
(1) A novel model architecture for debasing video QA
(2) Separate out the spurious relations with confounders
(3) Detailed explanations for the algorithms designs, and implementation details
(4) Good results on various benchmarks

Weakness:
(1) Nearly Four-times additional computational cost during training due to unanswerable Xq, Xv. in the original network f and the confounder encoder g; Good to include training times
(2) Good to have ablation study on the two component of Z (visual and textual)separately, currently only have them as a whole.
(3) Good to have ablation study on the number of the confounders



**Summary Of The Paper:**

The paper proposed a debasing approach to better video QA performance that utilizes learnt confounders to discover and fix spurious correlations. The confounders include both data-agnostic item, i.e. a learnable base (Z) and a data-specific item that depends on the video and textual input (Z'). The model jointly learns to output the right answer with the original input (X) and output the same prediction with created unanswerable input (X') using the confounders.

**Summary Of The Review:**

The paper introduces an approach to using confounders to separate out spurious relations in order to debasing video QA. The idea is novel and obtains good results, however with large additional computational cost.

---

> ### Author Response · Authors · 2022-11-14
> **Response for Reviewer EnYB**
>
> We appreciate **Reviewer EnYB** for the constructive comments and support for our motivation and solution to the problem. We will address your concerns, hoping for more vigorous support for our paper.
>
> ---
>
> **Concern 1:**
>
> Nearly four-times additional computation cost during training. Good to include training times.
>
> **Answer:**
>
> Great question! The proposed method causes about 50% computational overhead  in our experiments on a single Tesla A100 GPU. Note that the proposed method with VoidQ and the baseline with standard training requires the exactly same computations in the video loader, video encoder, text loader, and text encoder. The computation overhead by our method occurs only after the feature extraction. We compute the modality-specific features (common computations) “once” and differently combine/fed them to the data encoder $f$ and confounder encoder $g$. This makes our implementation efficient.
>
> ---
>
> **Concern 2:**
>
> Good to have an ablation study on two components of $Z$ (visual and textual) separately.
>
> **Answer:**
>
> |  | $Z$ | $Z$ (text only) | $Z$ (video only) |
> | --- | :---: | :---: | :---: |
> | MSVD | 46.4 | 45.2 | 44.5 |
>
> As shown in the table, the performance slightly decreases by 0.8% when only using the text confounder queries. On the other hand, the performance decreases by 1.9% when only using the video confounder queries. This indicates that the dataset has a stronger text bias than video. This result has been added to the updated supplement.
>
> ---
>
> **Concern 3:**
>
> Good to have ablation study on the number of the confounders.
>
> **Answer:**
>
> We provide the result of an ablation study on the number of confounders below:
>
> |  | 4 | 8 | 16 | 32 | 64 | 128 | 256 | 512 | 1024 |
> | --- | :---: | :---: | :---: | :---: | :---: | :---: | :---: | :---: | :---: |
> | MSVD | 44.8 | 45.0 | 45.7 | 46.3 | 46.4 | 46.4 | 46.2 | 46.1 | 45.4 |
>
> As shown in the table, the model performs best when the number of confounders is near 64 or 128. In our experiments, we used 128 confounder queries. We also added this experiment in the updated supplement.

---

> > ### Author Response · Authors · 2022-12-08
> > **A gentle reminder for Reviewer EnYB**
> >
> > Dear **Reviewer EnYB**, thank you again for your effort in reviewing our paper.
> >
> > The author-reviewer discussion will be closed soon.
> > Through rebuttal, we have addressed all your concerns, and we believe that our responses have answered your suggestions and questions.
> > Please let me know if you have further questions or concerns.
> > We look forward to your feedback and discussion.
> >
> > Sincerely, Authors

---

### Official Review · Reviewer_LFSv · 2022-10-24

**Confidence:** 3
**Clarity, Quality, Novelty And Reproducibility:** 1. Clarity
**Correctness:** 3
**Technical Novelty And Significance:** 3
**Empirical Novelty And Significance:** Not applicable
**Recommendation:** 3

**Strength And Weaknesses:**

Strengths:
1. A strength of this approach is that it appeals to a well founded base vis a vis the causal methods.
2. I'm not sure how novel the larger approach is, but applying this to VideoQA in order to debias the question and the video is a good idea that I have not seen before.

Weaknesses:
1. Boy oh boy is this hard to read. The intro section has three different summaries. It's not until page 6 that the authors finish explaining how their method works, and it starts on page 3. Figure 3 does not help clarify much. In fact, I think it made my understanding even worse. Why do you even need Figure 2 showing the graphical interpretation of the do calculus? In fact, why do you need most of the causal inference preliminaries? A lot of what the authors write is fairly basic and can be summarized / pointed to in other works. Figure 4 is super confusing. Why are there arrows going in both directions? I have rarely, if ever, seen a NN diagram that operates like that. The notation used to explain this method is tremendously difficult to read. I *strongly* suggest improving this because this paper is just about unreadable given how many times it takes to go through and properly glean what's going on.
2. That's not even to mention that it would fail for anyone who isn't looking on an interface w color or to anyone who is color blind to blue/red. You should use color only to highlight, not for something with real purpose and meaning.

**Summary Of The Paper:**

This paper tries to fix bias in VideoQA models through causally approaches. The authors noticed that VideoQA models tend to just learn the dataset statistics ("How many ..." --> "2") and look to rectify that by forcing the model to answer questions that don't make sense for the input in order to disrupt the spurious correlations. The goal here is to enforce that the model only takes advantage of causal relations between the question and the video and makes use of both modalities.

They go on to run experiments on three benchmark VideoQA datasets and show favorable results, as well as apply GradCAM to assess what parts of the modalities are being taken into account.

**Summary Of The Review:**

My review of this paper is overwhelmed by how hard it is to grok the idea. This doesn't pass the bar and would take great revision to fix it. I do actually think it's possible in this round, but it's going to take a wholesale rewrite of the first 2/3. I strongly suspect the paper will be 100x better after doing that.

---

> ### Author Response · Authors · 2022-11-14
> **Response for Reviewer LFSv**
>
> We appreciate **Reviewer LFSv** for the acknowledgement that this paper has “a good idea with good empirical and qualitative evaluation.” We will address your concerns, hoping for more vigorous support for our paper.
>
> ---
>
> **Concern 1:**
>
> “Clarity: This paper gets a 0 for clarity. Please make this easier to read. It’s very challenging at the moment.”
>
> **Answer:**
>
> “Boy oh boy is this hard to read.”
>
> - We respectfully disagree with your comment "this paper is just about unreadable given how many times it takes to go through and properly glean what's going on". As **Reviewer aBdW** said “The paper is **well written** and **easy to follow**.”,  we believe that our paper including figures is not difficult to follow overall. Concretely, we introduce the basic concepts of causal inference for readers who are not familiar with causal inference.
>
> “Figure 4 is super confusing. Why are there arrows going in both directions? I have rarely, if ever, seen a NN diagram that operates like that.”
>
> - Also, Figure 4 does NOT illustrate a neural network diagram. It is the objective functions with the proposed unanswerable questions, and bidirectional arrows mean calculating the loss between ground truth $Y$ or $Y'$ and the predictions $\hat{Y}$s.
>
> “Figure 3 does not help clarify much. In fact, I think it made my understanding even worse.”
>
> - We’d like to clarify it if you have specific questions and update our manuscript accordingly if necessary.
>
> ---
>
> **Concern 2 :**
>
> “ That's not even to mention that it would fail for anyone who isn't looking on an interface w color or to anyone who is color blind to blue/red. You should use color only to highlight, not for something with real purpose and meaning.”
>
> **Answer :**
>
> Further, as **Reviewer EnYB** said, we present detailed explanations for the algorithm designs and implementation details, by highlighting the notation with color to alleviate the notation confusion for the convenience of readers. Although we additionally explained the meaning of colors to help readers, we believe that the readers can understand without the color information. As suggested, we updated our manuscript for the convenience of the color blind.

---

> > ### Author Response · Authors · 2022-12-08
> > **A gentle reminder for Reviewer LFSv**
> >
> > Dear **Reviewer LFSv**, thank you again for your effort in reviewing our paper.
> >
> > The author-reviewer discussion will be closed soon.
> > Through rebuttal, we have addressed all your concerns, and we believe that our responses have answered your suggestions and questions.
> > Please let me know if you have further questions or concerns.
> > We look forward to your feedback and discussion.
> >
> > Sincerely, Authors

---

### Official Review · Reviewer_aBdW · 2022-10-25

**Confidence:** 4
**Correctness:** 3
**Technical Novelty And Significance:** 2
**Empirical Novelty And Significance:** 2
**Recommendation:** 5

**Clarity, Quality, Novelty And Reproducibility:**

This paper is technically sound but similar ideas have been used in related tasks.

**Strength And Weaknesses:**

1. Strengths:

+ The paper is well written and easy to follow. Although similar works have been done in VQA, not much work on debiasing and robustness has been done in VideoQA settings.
+ The proposed method achieves state-of-the-art performance on major VideoQA datasets.

2. Weaknesses:

+ Similar studies have been done in VQA and image captioning [1, 2]. The idea of sampling “unanswerable” QA pairs to detect the shortcuts that VQA models exploit when answering questions has been studied. The proposed method would be more interesting if it focuses on identifying biases that are unique to video data.

[1] Wen, Zhiquan, et al. "Debiased Visual Question Answering from Feature and Sample Perspectives." Advances in Neural Information Processing Systems 34 (2021): 3784-3796.
[2] Yang, Xu, Hanwang Zhang, and Jianfei Cai. "Deconfounded image captioning: A causal retrospect." IEEE Transactions on Pattern Analysis and Machine Intelligence (2021).

+ It is unclear what biases are detected and reduced by the proposed method. Examples of only “how many” question type do not seem convincing enough.

+ The explanation of how the authors detect visual biases and linguistic biases could be clearer. They also should provide more detailed analysis of how reducing the negative impacts of these biases impacts the overall performance.

+ How do the authors make sure the newly created pairs of video-question are unanswerable? What if questions are general-purpose such as “what is in the video?” or “what action is performed in the video?”?

+ I have a doubt about the results of TGIF-QA dataset as if you train VideoQA models on each question type one by one, what biases do you detect? What makes the improvements so large? What about the Count questions? I do not see it in Table 1.

**Summary Of The Paper:**

The paper studies VideoQA through the lenses of causality where it attempts to break the spurious correlations caused by biases (e.g., linguistic biases, visual biases etc.) when predicting answers. It proposes to identify these biases by forcing VideoQA models to respond to unanswerable questions obtained by pairing videos and question-answer pairs of different random samples. The proposed method demonstrates its effectiveness on several VideoQA datasets.

**Summary Of The Review:**

The paper has contributions to VideoQA task specifically but similar ideas have been used in related tasks.

---

> ### Author Response · Authors · 2022-11-14
> **Response for Reviewer aBdW [3/3]**
>
> **Concern 6:**
>
> Results of TGIF-Count are not reported.
>
> **Answer:**
>
> We did not experiment on the counting task for the TGIF-QA dataset, which is a “regression” task, since the proposed method is designed for classification tasks following previous works [5, 6].  In detail, the proposed causal intervention method is a way to compute a debiased logit $\hat{Y}_f = P(Y|do(X))$ instead of $P(Y|X)$ for the classification. On the other hand, a regression model predicts the output value $\hat{Y}$ instead of the probability of output categories as in the classification task. In other words, the proposed causal intervention method needs minor modification to handle regression problems. Although no previous debiasing methods demonstrated their methods for regression tasks, extending debiasing methods to regression tasks is an interesting direction for future research.
>
> [5] Nan et al., “Interventional Video Grounding with Dual Contrastive Learning”, CVPR 2021.
>
> [6] Zhang et al., “Causal Intervention for Weakly-Supervised Semantic Segmentation”, NeurIPS 2021.

---

> > ### Author Response · Authors · 2022-12-08
> > **A gentle reminder for Reviewer aBdW**
> >
> > Dear **Reviewer aBdW**, thank you again for your effort in reviewing our paper.
> >
> > The author-reviewer discussion will be closed soon.
> > Through rebuttal, we have addressed all your concerns, and we believe that our responses have answered your suggestions and questions.
> > Please let me know if you have further questions or concerns.
> > We look forward to your feedback and discussion.
> >
> > Sincerely, Authors

---

> ### Author Response · Authors · 2022-11-14
> **Response for Reviwer aBdW [2/3]**
>
> **Concern 4:**
>
> What makes the improvements so large on **TGIF-QA**?
>
> **Answer:**
>
> All the baselines in Table 1 adopted various backbone models and different pipelines for VideoQA. For example, HGQA [3] adopted 3D CNN for video encoder and BERT for text encoder whereas VIOLET [4] used cross-modal transformer architecture for video & text encoder. VIOLET is the overall second best model and it achieved 92.5 and 95.7 in TGIF-Action and TGIF-Transition. We also adopted a similar cross-modal transformer encoder as the backbone model like VIOLET and the performance is improved by a margin of 1% and 1.6% in TGIF-Action and TGIF-Transition compared to VIOLET. In sum, we used suitable backbone networks along with our “learned” confounder queries and our framework properly suppresses the bias in TGIF-QA dataset.
>
> [3] Xiao et al., “Video as Conditional Graph Hierarchy for Multi-Granular Question Answering”, AAAI 2022.
>
> [4] Fu et al., "VIOLET: End-to-End Video-Language Transformers with Masked Visual-token Modeling”, arXiv 2021.
>
> ---
>
> **Concern 5:**
>
> It is unclear what biases are detected and reduced by the proposed method. Examples of only “how many” question type do not seem convincing enough. Also, what biases are detected as for each question type in TGIF-QA?
>
> **Answer:**
>
> We updated the supplement with more analyses of what biases are detected and reduced by the proposed method. In Section E of the supplement, Figure 9 illustrates how models’ predictions are corrected via causal intervention $P(Y|do(X))$. We discuss the detected biases for three representative question types in TGIF.
>
> - **TGIF-Action.** As shown in Figure 9(a), the model tends to predict 'shake head' without considering the visual context before the causal intervention. On the other hand, the prediction is corrected to 'rub something with fingers’ after the causal intervention. We believe that this case is biased to text since the 'shake head' co-occurs 141 times more than 'rub' with the word 'man' in the question. Here, the word 'man' serves as the confounder inducing the text bias.
> - **TGIF-Transition.** Figure 9(b) shows the text bias. In this case, the 'smile' & 'man' pair co-occurs 169 times more than 'dump' & 'man' pair, which leads the model to predict 'smile' only considering the text. However, after the causal intervention with $P(Y|do(X))$, the model predicts the answer correctly.
> - **TGIF-Frame.** Figure 9(c) illustrates the video-biased case. $P(Y|X)$ is likely to focus on the visually salient object 'cat' without considering the question. By applying causal intervention, the video bias is alleviated and the prediction is corrected to 'paw' from 'cat' considering both the video and question.
>
> In the main paper, using variants of GradCAM, we have investigated which words in a question or which parts in a video are taken into account by confounder queries to debias the predictions of VideoQA model with GradCAM $\nabla_{(X;\tilde{Z})} \hat{Y}^t_f$ and Counterfactual GradCAM $\nabla_{(X;\tilde{Z})}^c \hat{Y}_f^t$ for MSVD. We also provided more detailed descriptions of both GradCAMs in Section B.3 of the revised supplement.
>
> For TGIF-QA, Figure 10(a) in the updated supplement shows the same QA pair with Figure 9(b), which is biased to the text. The word 'man' is strongly highlighted by counterfactual GradCAM $\nabla_{(X;\tilde{Z})}^c \hat{Y}^t_f $ implying it negatively influences to predict the correct answer. On the other hand, GradCAM $\nabla_{(X;\tilde{Z})} \hat{Y}^t_f$ focuses on the word 'bucket' to output a correct answer 'dump ice water on himself'. This is consistent with our observation that the word 'man' is considered as the text confounder hindering the model from predicting correctly. In Figure 10(b), the video-biased QA pair come from Figure 9(c), counterfactual GradCAM $\nabla_{(X;\tilde{Z})}^c \hat{Y}^t_f$ shows that the object 'cat/kitten' which is visually salient object in the video hinders the model to predict the proper answer. However, GradCAM $\nabla_{(X;\tilde{Z})} \hat{Y}^t_f$ focuses on the object 'paw' in the video so the model correctly predicts the answer. This indicates that the object 'cat' serves as the video confounder in the video-biased sample.

---

> ### Author Response · Authors · 2022-11-14
> **Response for Reviewer aBdW [1/3]**
>
> We appreciate **Reviewer aBdW** for the insightful and constructive comments. We will address your concerns, hoping for more vigorous support for our paper.
>
> ---
>
> **Concern 1:**
>
> Lack of novelty.
>
> **Answer:**
>
> As **Reviewer H5yq** said, our contributions also include that our framework can alleviate the effect of confounders even if the confounders are not defined in advance by introducing “learnable” confounder queries. Most previous works studied debiasing with “predefined” confounders such as an object set and a set of verbs, which is NOT suitable for real-world settings. On the other hand, we introduce confounder query tokens and train them with unanswerable questions to separate out the bias in the dataset. To the best of our knowledge, this is the first attempt to introduce and train the “un-predefined” (or learnable) confounders in the literature of VideoQA.
>
> ---
>
> **Concern 2:**
>
> Explanation of detecting bias could be clearer.
>
> **Answer:**
>
> We refer to Figure 4 of the main paper for a more detailed explanation. As depicted in the figure, we consider two original QA pairs, $X = (x_q, x_v)$ with label $Y$ and $X^\prime = (x^\prime_q, x^\prime_v)$ with label $Y^\prime$, and they are as follows:
> > $x_q$ : “What is the lion doing?”
> $x_v$ : A video in which lion is resting
> $Y$ : “rest”
> $x^\prime_q$: “What is a woman diving in?”
> $x^\prime_v$: A video in which woman is diving into a river
> $Y^\prime$ : “river”
>
> These two QA pairs are mixed across samples to form $X_q = (x_q, x^\prime_v)$ and $X_v = (x^\prime_q, x_v)$. For example, the mixed QA pair $X_q= (x_q, x_v')$ in yellow boxes in Figure 4, is an unanswerable pair with the question “What is the lion doing?” and the video in which a woman is diving into a river. Then, given $X_q$ as an input QA pair, the model is forced to predict an answer $Y$ (”rest”) and $Y^\prime$ (”river”). Since the video of a woman diving does not provide any information to answer the question “What is the lion doing?”, the model ends up relying solely on the question to predict the answer $Y$ (”rest”). Therefore, the model captures the bias existing in the question, resulting in the model always predicting “rest” as an answer without considering input videos. To further force the biased reasoning process, we adopt the GCE loss between $\hat{Y}^{(q,q)}_{g}$ and $Y$ in Figure 4.
>
> Similarly, the model relies on the input video where a woman is diving into the river when asked to predict $Y^\prime$ (”river”) since the question “What is the lion doing?” does not provide any information to predict the answer “river”, therefore it encourages the model to capture bias existing in the video. Similarly GCE is adopted between $\hat{Y}^{(q,v)}_g$ and $Y^\prime$ in Figure 4.
>
> The same mechanism is also applied to the other pair $X_v = (x'_q, x_v)$ in purple boxes in Figure 4. Similarly, the model is forced to predict $Y$ and $Y^\prime$ as an answer given an unanswerable QA pair, therefore capturing bias existing in the video and text respectively. All of these four losses are added to result in the final loss $\mathcal{L}_\text{confounder}$, which forces confounder queries $Z$ and a confounder encoder $g$ to capture the bias.
>
> ---
>
> **Concern 3:**
>
> How do the authors make sure the newly created pairs of video-question are unanswerable?
>
> **Answer:**
>
> Great point! We here provide additional analysis to check whether newly-created pairs are unanswerable or not. We randomly sample 100 instances from TGIF-Action (MCQA) and MSRVTT (OEQA) respectively to show the statistics of unanswerable QA pairs. For each QA pair, we manually classify it into three categories: (A) Unanswerable QA pair, (B) Wrong answerable QA pair (answerable but inducing the wrong answer), and (C) answerable QA pair. The statistics of each dataset are as follows:
>
> |  | TGIF-Action (MCQA) | MSRVTT (OEQA) |
> | --- | :---: | :---: |
> | (A) Unanswerable | 86% | 80% |
> | (B) Wrong answerable | 7% | 15% |
> | (C) Answerable | 7% | 5% |
>
> Since the model can also learn the bias when it is forced to predict the wrong answer, we consider case (B) as (A), either. Then, the proportions of answerable question/answer pairs are only 7% and 5% in MCQA and OEQA, respectively. We believe that the portion of answerable question/answer pairs is sufficiently small and the bias by GCE loss ensures that our confounder queries learn biases as in the literature of [1, 2].
>
> [1] Nam et al., “Learning from Failure: Training Debiased Classifier from Biased Classifier”, NeurIPS 2020.
>
> [2] Lee et al., “Learning Debiased Representation via Disentangled Feature Augmentation”, NeurIPS 2021 (Oral).

---

### Author Response · Authors · 2022-11-19
**A Gentle Reminder for Reviewers**

We sincerely appreciate for constructive comments and your time in reviewing our paper. We have responded to the questions raised by reviewers and uploaded the revised version of our manuscript including the supplement. The major changes in our manuscript are summarized below:

---

1. Further analyses of what biases are detected and reduced on TGIF-QA (**Reviewer aBdW**).
    - We have provided detailed descriptions of GradCAM and Counterfactual GradCAM in Section B.3 of the supplement. In Section E of the supplement, we also have discussed the detected biases for three representative question types in TGIF-QA.
2. Improve readability (**Reviewer LFSv**).
    - We have updated Section 3.3 of the main paper to improve the readability for the convenience of the color blind.
3. More ablation studies for confounder queries $Z$. (**Reviewer EnYB**).
    - We have provided the ablation study for the type of confounder queries $Z$ in Section F.1 of the supplement.
    - We have provided the ablation study for the number of confounder queries $Z$ in Section F.2 of the supplement.

---

We believe that our responses and updated manuscript have addressed your suggestions and questions. Please go over our responses and let us know if you have any further questions. Thank you.

---

### Decision · Program_Chairs · 2023-01-20

**Decision:**

Reject

**Justification For Why Not Higher Score:**

1. Unconvincing experiments.
2. The quality of the presentation does not meet the standard of ICLR.

**Justification For Why Not Lower Score:**

NA

**Metareview: Summary, Strengths And Weaknesses:**

This paper was reviewed by four experts in the field and received a mixed score. The main concerns are the limited novelty, unconvincing experiments, and lack of clarity. The authors did a good job of rebuttal and addressed many of the concerns. However, the reviewers (including all positive ones) still feel that more work is needed to get it to the best version. AC also agrees that this work can be much stronger with additional experiments. While this paper clearly has merit, the decision is not to recommend acceptance. The authors are encouraged to consider the reviewers' comments when revising the paper for submission elsewhere.